# A *nop56* Zebrafish Loss-of-Function Model Exhibits a Severe Neurodegenerative Phenotype

**DOI:** 10.3390/biomedicines10081814

**Published:** 2022-07-28

**Authors:** Ana Quelle-Regaldie, Mónica Folgueira, Julián Yáñez, Daniel Sobrido-Cameán, Anabel Alba-González, Antón Barreiro-Iglesias, María-Jesús Sobrido, Laura Sánchez

**Affiliations:** 1Department of Zoology, Genetics and Physical Anthropology, Faculty of Veterinary Science, University of Santiago de Compostela, 27002 Lugo, Spain; ana.quelle@usc.es; 2Instituto de Investigación Biomédica de A Coruña (INIBIC), Servicio Galego de Saúde, 15008 La Coruña, Spain; ssobrido@gmail.com; 3Department of Biology, Faculty of Sciences, University of A Coruña, 15008 La Coruña, Spain; m.folgueira@udc.es (M.F.); julian.yanez@udc.es (J.Y.); anabel.albag@udc.es (A.A.-G.); 4Centro de Investigaciones Científicas Avanzadas (CICA), University of A Coruña, 15008 La Coruña, Spain; 5Department of Functional Biology, Faculty of Biology, Centro de Investigaciones Biológicas CIBUS, University of Santiago de Compostela, 15706 Santiago de Compostela, Spain; ds918@cam.ac.uk (D.S.-C.); anton.barreiro@usc.es (A.B.-I.)

**Keywords:** zebrafish, neurodegeneration, animal models, genetic edition

## Abstract

NOP56 belongs to a C/D box small nucleolar ribonucleoprotein complex that is in charge of cleavage and modification of precursor ribosomal RNAs and assembly of the 60S ribosomal subunit. An intronic expansion in *NOP56* gene causes Spinocerebellar Ataxia type 36, a typical late-onset autosomal dominant ataxia. Although vertebrate animal models were created for the intronic expansion, none was studied for the loss of function of *NOP56*. We studied a zebrafish loss-of-function model of the *nop56* gene which shows 70% homology with the human gene. We observed a severe neurodegenerative phenotype in *nop56* mutants, characterized mainly by absence of cerebellum, reduced numbers of spinal cord neurons, high levels of apoptosis in the central nervous system (CNS) and impaired movement, resulting in death before 7 days post-fertilization. Gene expression of genes related to C/D box complex, balance and CNS development was impaired in *nop56* mutants. In our study, we characterized the first NOP56 loss-of-function vertebrate model, which is important to further understand the role of NOP56 in CNS function and development.

## 1. Introduction

NOP56 is part of a core protein complex, along with NOP58, fibrillarin (FBL) and a 15.5 kD RNA-binding protein, of the C/D box small nucleolar ribonucleoprotein (snoRNP), which is in charge of cleavage and modification of precursor ribosomal RNAs (pre-rRNAs) and assembly of the 60S ribosomal subunit [1,2,3] Specifically, this C/D box snoRNP introduces a direct 2′-O-ribose methylation in specific residues in pre-rRNA and is involved in the endonucleolytic cleavages of the 35S rRNA primary transcript [4]. This C/D complex is highly conserved throughout evolution from archaea to humans [3,5,6,7,8,9].

NOP56 is decisive for the final maturation and activation of a catalytically active C/D box snoRNP. Small nucleolar RNAs (snoRNAs) control protein levels of NOP56 depending on the availability of C/D box snoRNP assembly factors [10]. Mutations in *NOP56* in yeast and human cells lead to altered ribosomal biogenesis, notably on the pathway of 25S/5.8S rRNA synthesis, produce defects in 60S subunit assembly and disrupt cell cycle progression [7,10].

C/D complex dysregulation has been associated with cancer, mainly by effects on snoRNAs, which show differential expression patterns in diverse types of human cancers and are able to alter cell transformation, tumorigenesis and metastasis [11,12,13]. *NOP56* was found to be a critical gene associated with Myc protooncogene transformation to produce Burkitt’s lymphomas [14]. *NOP58* and fibrillarin are also associated with hepatocellular carcinoma and breast cancer, respectively [15,16,17].

Spinocerebellar Ataxia type 36 (SCA36), also known as “Costa da Morte ataxia” [18] and “Asidan ataxia” [19], is caused by a heterozygous expansion of the “GGCCTG” repeats within *NOP56* intron (referred as “SCA36 expansion” throughout the text). SCA36 is a type of late-onset autosomal dominant ataxia characterized by cerebellar ataxia, sensory hearing loss and discrete motor neuron impairment (tongue atrophy with denervation, discrete pyramidal signs) [18,19]. Magnetic resonance imaging (MRI) of SCA36 brains shows cerebellar atrophy, while neuropathological studies revealed Purkinje and motor neuron loss [20,21]. As no reduction in NOP56 protein levels was found in affected human lymphoblastoids and RNA foci were detected, SCA36 is thought to be caused by a toxic mRNA gain of function or a combination of toxic gain of function and loss of function [19]. Although symptoms are milder, the hexonucleotide repeated mutation in SCA36 is similar to the *C9ORF72* gene mutation that causes Amyotrophic Lateral Sclerosis (ALS). *C9ORF72* mutation of ALS consists of an intronic repetition of the “GGGGCC” expansion and has been proposed to function as a gain of toxic effect on RNA metabolism [22,23]. However, some studies suggest that ALS could be caused by a combination of gain-of-function and loss-of-function mechanisms [22,24,25]. As the mechanisms of ALS have been much more deeply studied than those of SCA36, it would be interesting to see what happens under loss of function of NOP56 in an animal model. Despite efforts to unravel how SCA36 repeats lead to disease, currently there is no vertebrate model with a similar stable mutation in their genome to that of human patients. In vertebrates, only transient models were generated in mouse. So far, most models of the nucleotide expansion observed in SCA36 have been developed in cell cultures [26,27,28]. *Nop56* transgenic models in Drosophila shows defects in optic lobe development, which is caused by inhibition of cell cycle progression [29]. Recently, McEachin et al. (2020) demonstrated that SCA36 expansion caused intronic retention in mRNA and repeat-associated non-AUG (RAN) translation that caused five different dipeptide repeat proteins (DPRs) in samples from SCA36 patients. In addition, injection of DNA constructs with the SCA36 expansion in mice results in inclusions of DPRs in the central nervous system (CNS), but not other hallmarks of SCA36 [28]. Another mouse model, generated by injection of constructs containing 62 repeats of the SCA36 expansion, shows pronounced locomotor defect, loss of cerebellar Purkinje cells, RNA foci and DPR inclusions in the brains [30].

Zebrafish (Danio rerio) has become an important model for the study of hereditary neurological disorders, as it has orthologs for 76% to 82% of human genes and its general organization of the CNS is highly similar to humans and many CNS regions relevant for human disease studies show molecular and structural homology in zebrafish [31]. Many zebrafish genetic models have been developed for neurological disorders, including ataxias [32,33,34,35,36,37] and ALS [38,39,40,41].

Given that a reduction in mRNA and protein levels was not studied in SCA36 pathogenesis, here we characterize a zebrafish loss-of-function model of the *nop56* gene, which shows 70% homology with the human gene. We observed an aberrant phenotype in homozygous *nop56* embryos that includes smaller size, malformations in the jaw, eye, inner ear and brain (e.g., absence of cerebellum and reduced number of spinal cord neurons), and impaired movement, resulting in death before 7 days post-fertilization (dpf). Homozygous mutants also showed high levels of apoptosis in the CNS. Transcription of *nop56* gene was reduced in homozygous mutants, while *nop58* and *fbl* were overexpressed, so rRNA processing seems to be impaired. *C9orf72*, *fus* and *tardbp* genes, related to Amyotrophic Lateral Sclerosis (ALS), also had reduced mRNA expression in *nop56* homozygous mutants. In addition, expression of *zpld1a* and *zpld1b* genes, related to the development of the cupula of the inner ear, which is responsible for balance, were significantly reduced. Our study provides the first NOP56 loss-of-function vertebrate model, which will be an important tool to further understand the role of NOP56 in CNS development and function.

## 2. Materials and Methods

### 2.1. Zebrafish Care and Maintenance

Zebrafish carrying the wild-type and mutant *nop56*^sa12582^ alleles [42] were obtained from the European Zebrafish Resource Center (EZRC). The *Nop56*^sa12582^ line was generated by the ENU method (N-ethyl-N-nitrosourea) [43], which caused a C > T change that resulted in a premature stop codon.

Zebrafish individuals were maintained in the fish facilities of the Department of Genetics of the University of Santiago de Compostela (code of ethical approval committee and for animal experiments: AE-LU-003, ES270280346401) at 28 °C with a photoperiod of 14 h of light and 10 h of darkness according to standardized protocols [44,45]. All experiments involving animals followed the guidelines of the European Community and Spanish Government on animal care and experimentation (Directive 2012-63-UE and RD 53/2013).

Heterozygous fish (*nop56*^+/−^) were crossed to obtain wild-type, heterozygous (*nop56*^+/−^) and homozygous embryos (*nop56*^−/−^). Embryos were obtained after spawning at the beginning of the light morning cycle. Embryos were kept at 28 °C in Petri dishes with E3 medium (5 mM NaCl, 0.17 mM KCl, 0.33 mM CaCl_2_ and 0.33 mM MgSO_4_).

### 2.2. Genotyping

DNA was extracted with *Chelex* 100 Resin (1422822, Bio-Rad; Hercules, CA, USA) and amplified by PCR with AmpliTaq Gold™ DNA Polymerase (Thermo Fisher Scientific; Waltham, MA, USA) in a thermal cycler with the primers specific for the mutation in *nop56*^sa12582^: F: 5′ TGGCGGAAGATTTGATTCTG 3′ and R: 5′ TTTCCACTCGACATTCATCG 3′. PCR products were detected using 1% agarose gels. Sequencing of the PCR products were performed using the capillary sequencer 3730xl DNA Analyzer (Thermo Fisher Scientific) by Sanger sequencing. The results were analyzed with GeneMapper^®^ Software (Thermo Fisher Scientific; Waltham, MA, USA) and aligned using the CodonCode Aligner software (Codon Code Corporation; Centerville, USA). The results indicated if the analyzed individuals possess the mutation of interest and whether they are homozygous or heterozygous. Finally, we analyzed if the segregation of the genotypes was Mendelian by using a chi-square test.

### 2.3. Characterization of General Morphology

Fish carrying wild-type and mutant *nop56* ^sa12582^ alleles were used for a phenotypic study between 24 and 120 h post-fertilization (hpf). For this, 446 embryos and larvae were dechorionated and anesthetized with 0.002% tricaine methanesulfonate (Ms-222, Sigma-Aldrich; Saint Louis, MO, USA). Photographs were taken every 24 h with a Nikon Ds-Ri1 camera coupled to an inverted fluorescence microscope (AZ100 Multizoom Nikon; Tokyo, Japan). Images were analyzed with ImageJ software (National Institutes of Health; Bethesda, MD, USA), annotating alterations in general morphology and measuring parameters such as total length of the body and size of the head, eye and otoliths. All fish were genotyped by the end of the experiments. For more detailed analysis, a few selected embryos and larvae were mounted in lateral and ventral views in 1% agarose in phosphate-buffered saline pH 7.4 (PBS) and imaged likewise at higher magnification. In this case, we performed a detailed analysis of malformations in the jaw, body midline, somites and notochord.

In addition, brains and eyes of adult wild-type and heterozygous fish were analyzed. Adult fish were euthanized by tricaine methanesulfonate overdose and fixed overnight in 4% paraformaldehyde (PFA) in PBS at 4 °C. After dissection of the brain and eyes, these were imaged under a stereomicroscope. ImageJ software was then used to measure the area of the brain and optic tectum.

### 2.4. Survival Analysis

Survival rate of each group (wild types, heterozygous and homozygous) was calculated. A Kaplan–Meier graph of survival between 1 and 7 days post-fertilization was generated and the average days of survival were calculated. All fish were genotyped after the experiments.

### 2.5. Expression Analysis

To perform an mRNA expression analysis, we used larvae between 72 and 120 hpf from a heterozygous incross. The resulting larvae were sectioned into two parts—head and tail. The heads were put in RNAlater (Sigma-Aldrich; Saint Louis, MO, USA). The tails were used to determine the genotype as previously indicated. After genotyping, pools of larval heads of each of the three groups (wild type, *nop56*^+/−^ and *nop56*^−/−^) were selected. We used 3 pools (20–25 heads of larvae in each pool) for each experimental group and age. Total RNA was extracted with TRIzol Reagent (Invitrogen; Waltham, MA, USA) and treated with RNase-Free DNase Set (Qiagen; Germantown, MD, USA). RNA was quantified using NanoDrop^®^ 2000 (Thermo Fisher Scientific). cDNA was generated with the kit Affinity Script cDNA Synthesis (Agilent; Santa Clara, CA, USA). RT-qPCR was carried out with the miRNA QRT-PCR Detection Kit (Agilent Santa Clara, CA, USA) and using the following primers specific for *nop56* and other genes of interest:

*nop56*-F: 5′ CCCACAAGTGTGTTTGGTGA 3′

*nop56*-R: 5′ CCTTCTGCAGTGGAAAGAGC 3′

*nop58*-F: 5′ GGACAGACCACAGCCAAGAA 3′

*nop58*-R: 5′ CGCCTGATCCCTTTCTCCTC 3′

*fbl*-F: 5′ ATCAGTGCAGCAGTAGACCG 3′

*fbl*-R: 5′ CCTTGGGCTGAATCCTGGTC 3′

*zpld1a*-F: 5′ ACCCGAAGTGATGAAACTCCC 3′

*zpld1a*-R: 5′ GCATTTAGCTGAACGCTGGG 3′

*zpld1b*-F: 5′ CATCAGGGGACTGAGCTGTT 3′

*zpld1b*-R: 5′ CTCCATGACGGCTGTTCAGT 3′

*c9orf72-*F: 5′ ACACTGTGCTCAACGACGAT 3′

*c9orf72-*R: 5′ AACTGCCGGTGGAGTCCTTA 3′

*tardbp*-F: 5′ CTTTGCAGATGACCAGGTTGC 3′

*tardbp*-R: 5′ CCTCCGAACCCATTCCCAAA 3′

*fus*-F: 5′ AGCAGAGTGGAGGTGGGTAT 3′

*fus-*R: 5′ GTTGTAACCTCCCTGTCCGT 3′

*cbln12-*F: 5′ GCCCACTAATTCAGGTTGCCT 3′

*cbln12-*R: 5′ CCCGTCTGTCAGTTTTCCCT 3′

*grid2-*F: 5′ GTCCCATCGAAGGAGGACGATA 3′

*grid2-*R: 5′ AGAGTCTGGATCTGCTCTGGT 3′

*ptf1a-*F: 5′ ACATGCCAATTCGGAACCCA 3′

*ptf1a-*R: 5′ CATTTGGAGATGGGGATCTTGTT 3′

*p21**-*F: 5′ CGCAAACAGACCAACATCAC 3′

*p21**-*R: 5′ ATGCAGCTCCAGACAGATGA 3′

In addition, primers sequences of 5′ETS, 18S, ITS1, ITS2 and *p53* were obtained from Bouffard et al., 2018 [46].

Each sample was run in triplicate and we used primers for beta-actin 2 as housekeeping gene, F: 5′ ACTTCACGCCGACTCAAACT 3′ R: 5′ ATCCTGAGTCAAGCGCCAAA 3′. Relative mRNA levels of each gene were normalized to the expression of the housekeeping gene through the 2^∆∆CT^ calculation method.

### 2.6. Acridine Orange

Apoptotic cells of wild-type, *nop56*^+/−^ and *nop56*^−/−^ embryos at 24–30 hpf were stained with a 3 μg/mL solution of acridine orange in E3 medium for 30 min. After that, the embryos were washed twice in E3 medium. Confocal photomicrographs were taken with a Leica TCS SPE confocal laser microscope (Leica Microsystems, Wetzlar, Germany) with the GFP filter set (excitation 473, emission 520) at 5 × magnification and always using the same parameters of laser intensity and photomultiplier gain. The images were analyzed with ImageJ software (National Institutes of Health; Bethesda, MD, USA). Pixels of the embryo were selected using the tracing tool and the area; the mean and the integrated density of each fish were measured. These measurements were combined in the formula of corrected total cell fluorescence (CTCF): CTCF = integrated density − (area of selected cell × mean fluorescence of background readings).

### 2.7. Immunofluorescence

Fish were euthanized by tricaine methanesulfonate overdose and then fixed with 4% paraformaldehyde (PFA) in phosphate-buffered saline (PBS; pH 7.4) for at least 2 h at room temperature.

For serotonin (5-HT) and zebrin immunofluorescence experiments, after washes in PBS, larvae were incubated in collagenase (2 mg/mL in PBS) for 25 min at room temperature and then in glycine (50 mM in PBS with 1% Triton X-100) for 10 min at room temperature. Then, the animals were incubated with *rabbit anti-5-HT* (Immunostar, Still Water, MN, USA; Cat#: 20080; dilution 1:2500) or *Zebrin II* (mouse anti-Aldolase C; kindly provided by Prof. Richard Hawkes: dilution 1:100) antibodies overnight at 4 °C. The larvae were rinsed in PBS with 1% Triton X-100 (1% PBST) and incubated overnight at 4 °C with a Cy3-conjugated goat anti-rabbit antibody (Millipore; Burlington, MA, USA; dilution 1:200) or a FITC-conjugated goat anti-mouse antibody (Millipore; dilution 1:100). Antibodies were always diluted in 1% PBST, 1% DMSO, 1% normal goat serum and 1% bovine serum albumin. Larvae were mounted with 70% glycerol in PBS. Confocal photomicrographs were taken with TCS-SP2 spectral confocal laser microscope (Leica Microsystems, Wetzlar, Germany). For the quantification of serotonergic neurons, the total number of 5-HT-ir interneurons located in the 4 spinal cord segments located at the level of the caudal fin was quantified manually using the cell counter plugin of the Fiji software (Image J; National Institutes of Health; Bethesda, MD, USA) going through the stack of confocal optical sections.

For alpha-tubulin and synaptic vesicle 2 (SV2) immunofluorescence, fish were transferred to 100% methanol and kept at −20 °C for at least 30 min. After rinsing in PBS with 0.5% Triton X-100 (0.5% PBST), fish were digested in proteinase K (Sigma, Aldrich; Saint Louis, MO, USA), fixed in PFA and incubated with normal goat serum in 0.5% PBST for 1 h. Then, fish were incubated with alpha-tubulin (Sigma-Aldrich; Saint Louis, USA; Cat#: T7451, dilution 1:250) or SV2 (DSHB; Cat#: AB2315387, dilution 1:250) primary antibodies diluted in 1% normal goat serum in PBST overnight at 4 °C. The next day, after washing in PBST, fish were incubated overnight with goat anti mouse antibodies conjugated to Alexa Fluor 568 (Invitrogen, Waltham, USA; A-11029 and A-11031) in 1% normal goat serum in PBST. After washes, some fish were counter stained with Sytox Green Nuclear Stain (Invitrogen, Waltham, MA, USA; Cat#: S7020; dilution 1:1000). Fish were mounted in 1% agarose in 80% glycerol and imaged using a laser scanning confocal microscope Nikon A1R (Nikon; Tokyo, Japan).

### 2.8. Behavioral Analysis

Embryos between 2 and 3 dpf were dechorionated with a pair of forceps and were subjected to touch-evoked escape response analysis. Behavior was recorded with a Nikon Ds-Ri1 camera coupled to an inverted fluorescence microscope (AZ100 Multizoom Nikon; Tokyo, Japan). Distance moved was measured under quantification mode of Zebralab software (Viewpoint).

Movement of 4 dpf zebrafish larvae was quantified with a Zebralab system composed of a Zebrabox (Viewpoint; Civrieux, France). The 4 dpf larvae were introduced in 96-well plates and total movement was measured in periods of 10 min during an hour alternating light and dark.

### 2.9. Morpholino Injection

*p53* morpholino injection was used to assess that the observed phenotype is not a consequence of upregulation of p53-mediated apoptosis caused by off-target effects. *p53* morpholino was designed and synthesized by Gene Tools (Philomath, OR, USA). Its sequence is: 5′GCGCCATTGCTTTGCAAGAATTG3′. It was injected into one-cell-stage embryos in a volume of 3–5 nL/embryo and at a concentration of 1 mM.

### 2.10. Statistical Analysis

Data analysis of the three groups (wild type, *nop56*^+/−^ and *nop56*^−/−^) and graphs were generated with GraphPad Prism (GraphPad; San Diego, CA, USA), using one-way ANOVA with Bonferroni test for multiple comparisons after performing the D’Agostino–Pearson normality test. Statistical significance was established at *p*-value < 0.05. Statistically significant tests are indicated with asterisks in the graphs.

## 3. Results

### 3.1. nop56^−/−^ Embryos Develop an Abnormal Phenotype

From 24 to 120 hpf, we confirmed the expected Mendelian distribution of genotypes from an incross between heterozygous using chi-square test, with 27.5% of wild type (*n* = 123), 48% of *nop56*^+/−^ (*n* = 213) and 24.5% of *nop56*^−/−^ (*n* = 110).

*Nop56^−/−^* individuals have a characteristic abnormal phenotype, showing a reduced body length (differences are statistically significant from 48 hpf onwards, *p*-value < 0.001) and a smaller head, otoliths and eyes (these 3 characteristics being statistically significant from 24 hpf onwards: *p*-value < 0.001) (Figure 1). In addition, after detailed analysis, we also observed abnormal eye morphology in the homozygous fish compared to wild type, including coloboma and midline malformations (Figure 2). We also observed other characteristics such as swollen yolk in 100% of the *nop56*^−/−^ embryos since 48 hpf (less than 2% of the wild type and *nop56*^+/−^ had swollen yolk). Cardiac edema was detected in 77% of the *nop56*^−/−^ embryos at 72 hpf, 88% at 96 hpf and in 100% at 120 hpf (less than 3% of the wild type and *nop56*^+/−^ had cardiac edema). Lordosis was first observed at 48 hpf in 40% of *nop56*^−/−^, at 120 hpf the number rises to 50% of the larvae (less than 5% of the wild type and *nop56*^+/−^ had this malformation at this developmental stage). *Nop56*^−/−^ embryos fail to hatch from the chorion, but this is not the cause of the curved body shape because when manually removed from the chorion, they keep their curved body shape. From 96 hpf, we observed absence of the swimming bladder in 100% of the *nop56*^−/−^ embryos (100% of the wild-type and *nop56*^+/−^ embryos had swimming bladder).

In addition, a more detailed analysis under a bright-field microscope showed additional malformations in homozygous fish compared to wild types by 3.5 dpf (Figure 2), mainly affecting the jaw, eye and brain. Homozygous fish had much smaller jaw and seemed to show abnormal patterning of the brain, with much smaller or absent cerebellum and smaller midbrain.

In order to analyze possible macroscopic differences in the brain of heterozygous individuals for the *nop56* gene in adult zebrafish, the dimensions of different regions of fixed brains were measured. Data analysis between wild-type individuals and heterozygotes carrying the mutation at six months of age, resulted in no significant differences observed in the sizes corresponding to different areas of the brain, including the cerebellum (not shown).

Finally, we analyzed survival of wild-type (*n* = 43), *nop56*^+/−^ (*n* = 75) and *nop56*^−/−^ (*n* = 44) zebrafish during their first week of life and plotted their survival curves using a Kaplan–Meier survival graph (Figure 1B). A log-rank statistical test comparing the individual group survival curves revealed that the *nop56*^−/−^ had a significantly shorter survival with decease start as soon as 4 dpf as compared to *nop56*^+/−^ and wild-type (*p* < 0.0001) animals during early development. Our results showed that 80% of the *nop56*^−/−^ larvae died before 6 dpf and all the *nop56*^−/−^ animals died by 7 dpf.

### 3.2. Malformations in the Central and Peripheral Nervous System of nop56 ^−/−^ Larvae

We quantified mRNA levels by RT-qPCR of three genes that are expressed in specific cell types of the zebrafish cerebellum: *ptf1a* and *grid2* (Purkinje cells) and *cbln12* (granule cells) [47,48]. We observed a significant reduction in *ptf1a*, *grid2* and *cbln12* in *nop56^−/−^* (*p*-value < 0.0001) (Figure 3). *grid2* and *cbln12* expression was observed to be also reduced in *nop56^+/−^* between 72 and 120 hpf in comparison with wild-type embryos (*p*-value < 0.0001) but not as strong as *nop56^−/−^* larvae.

We analyzed in more detail the central and peripheral nervous system in 3.5 dpf wild type, *nop56*^+/−^ and *nop56*^−/−^ by using alpha-tubulin and synaptic vesicles 2 (SV2). In the head, we observed abnormal innervation of the jaw (Figure 4). In the brain, we observed smaller optic tectum and smaller or absent cerebellum (Figure 4). Zebrin II immunofluorescence experiments revealed a complete lack of cerebellar Purkinje cells in 4 dpf *nop56*^−/−^ animals as compared to wild-type or heterozygous fish (Figure 5).

### 3.3. Neuromuscular Junction and 5-HT Spinal Cord Interneurons

We next analyzed neuromuscular junction integrity. For this, we used alpha-tubulin to label the whole trunk innervation and SV2 as a presynaptic marker. Both markers revealed abnormal innervation of the trunk myomeres in *nop56*^−/−^ compared with wild types by 3.5 dpf (Figure 6). Alpha-tubulin shows that axons of primary motor neurons are thicker and less branched in *nop56*^−/−^ than in wild types (Figure 6 and Appendix A). SV2 also shows less branching of the axons innervating the myomeres, as well as differences in myosepta staining.

We also observed that the number of 5-HT-ir neurons was significantly reduced in the spinal cord of 4 dpf *nop56*^−/−^ larvae (one-way ANOVA, *p* < 0.0001) as compared to control wild-type animals (Figure 7). No significant differences were observed between control wild-type and *nop56*^+/−^ animals.

### 3.4. nop56^−/−^ Larvae Have Increased Apoptosis Mainly in the CNS

Acridine orange was used for in vivo staining of cellular apoptosis. An increase in apoptosis was seen in the *nop56*^−/−^ embryos (*p*-value < 0.0001), mainly in the eye, brain and spinal cord (Figure 8).

We examined mRNA expression by RT-qPCR of *p21* and *p53* genes that are related to apoptosis and cell cycle control [49]. We found both were overexpressed in *nop56*^−/−^ larvae (*p*-value < 0.0001). The microinjection of a *p53* translation block morfolino at one-cell-stage did not rescue the *nop56*^−/−^ malformations (not shown) even when the expression of *p53* was similar to wild type when measured at 72 hpf (Figure 9). The reduction in *p53* mRNA concentrations can be explained by the fact that it was observed that sometimes when a translation block morpholino binds to an mRNA, its secondary structure suffer changes, altering the availability of mRNA for nucleotyc degradation [50].

### 3.5. nop56 Knockout Caused Impaired Motor Function

At 48 hpf, *nop56*^−/−^ embryos were observed to develop a balance defect characterized with failure to maintain an upright position during swimming with the presence of trembling (Appendix A).

Touch-evoked response at 48 hpf was measured and observed to be impaired in *nop56*^−/−^ embryos. Moreover, when locomotion was measured at 96 hpf, that *nop56*^−/−^ larvae did not have the ability to swim was observed (Figure 10), which is consistent with our observation of abnormalities in neuromuscular junctions and spinal cord interneurons.

### 3.6. Reduced Expression of the zpld1 Genes from the Cupula of the Inner Ear

*ZPLD1* (zona pellucida-like domain 1) is a gene expressed in the cupula, a gelatinous membrane overlying the crista ampullaris of the semicircular canal in the inner ear, which is important for sensing rotation of the head and critical for normal balance [51]. In zebrafish, there are two paralogous genes: *zpld1a* and *zpld1b*. Since SCA36 patients have hearing loss, and *nop56*^−/−^ zebrafish embryos developed a balance defect, we performed an expression analysis of these genes in our loss-of-function model. The expression of *zpld1a* and *zpld1b* in *nop56*^−/−^ was significantly reduced in homozygous mutants as compared to wild-type and heterozygous larvae (*p*-value < 0.0001) (Figure 11). *zpld1b* was also significantly reduced in *nop56*^+/−^ embryos at 120 hpf, but not as strong as npc1^−/−^ expression. 

### 3.7. nop56 Disruption Causes Overexpression in nop58 and fbl Related Genes and Impaired rRNA Processing

NOP56 is a protein required for the assembly of the 60S ribosomal subunit. Together with NOP58 and FBL, they function as core proteins to form the C/D box small nucleolar ribonucleoprotein that modifies and processes ribosomal RNAs (see introduction). For this reason, we examined the mRNA expression of *nop56* gene and associated genes *nop58* and *fbl* by RT-qPCR (Figure 12). We observed significant differences in *nop56* expression between wild type, *nop56*^+/−^ and *nop56*^−/−^ embryos (*p*-value < 0.0001). nop56^−/−^ embryos have a reduced expression of *nop56* and *nop56*^+/−^ embryos have an intermediate expression between *nop56*^−/−^ and wild type. *Nop58* expression levels were increased in *nop56*^−/−^ larvae respect to wild type and *nop56*^+/−^ (*p*-value < 0.0001). fbl expression levels were also increased at 120 hpf in *nop56*^+/−^ and *nop56*^−/−^ embryos respect to wild-type animals (*p*-value < 0.0001).

In addition, as NOP56 participates in rRNAs processing, we examined by RT-qPCR the rRNA expression at 120 hpf of the 5′ externally transcribed sequence (5ETS) and internally transcribed sequences (ITS1 and ITS2) of the 47S intermediate rRNA, whose transcription is implicated in the first step of ribosome biogenesis. Moreover, we quantified by RT-qPCR the expression at 120 hpf of 18S rRNA, which is generated after processing of the 47S one. We obtained similar results as in a zebrafish knockout model of *fbl* gene [46]. 18S rRNA levels were significantly lower in nop56^−/−^ in comparison with nop56^+/−^ and wild type (*p*-value < 0.0001), which means that rRNA processing is impaired. 5′ETS and ITS2 expression did not differ between the three groups, although we detected a surprising overexpression of ITS1 (*p*-value < 0.0001) (Figure 13). This indicates that 47S rRNA transcription was not affected but subsequent processing was indeed affected.

### 3.8. Expression of Genes Related to ALS Is Also Affected in the nop56 Knockout Embryos

Mutations in autophagy regulator protein C9ORF72 and in DNA/RNA binding proteins TDP-43 and FUS are associated with Amyotrophic Lateral Sclerosis (ALS) disease. A hexanucleotide (GGGGCC) expansion in a noncoding region of *C9ORF72* gene is the main cause of ALS and Frontotemportal Dementia (FTD). This expansion generates RNA foci, RNA/DNA G-quadruplexes and dipeptide repeat proteins (DPRs) that cause neurotoxicity [52]. *TARDBP* encodes the protein TDP-43, which regulates transcription, alternative splicing of mRNA and non-homologous end joining (NHEJ) repair of DNA in motor neurons [53]. FUS (Fused in Sarcoma) is involved in RNA metabolism (transcription, splicing and export to cytoplasm) and DNA repair of double strand breaks through DNA damage response [54]. It has been shown that *FUS* loss of function causes impairment of proper DNA damage response leading to neurodegeneration and formation of aggregates [55]. Expression assay of *c9orf72*, *tardbp* and *fus* (Figure 14) revealed a significant reduction in the expression of these genes in the *nop56*^−/−^ embryos (*c9orf72 p*-value < 0.001, *tardbp p*-value < 0.001, *fus p*-value < 0.0001).

## 4. Discussion

In this work, we characterized a zebrafish loss-of-function model of the *nop56* gene. Homozygous mutants *nop56*^+/−^ showed a severe neurodegenerative phenotype characterized by, among other features, absence of cerebellum, reduced numbers of spinal cord neurons and impaired movement. Apoptosis pathway was increased in the CNS causing death before 7 days post-fertilization (dpf). *Nop58* and *fbl* genes were overexpressed, while rRNA processing seems to be impaired. Other genes with reduced expression in nop56^−/−^ were *c9orf72*, *fus, tardbp* (genes related to ALS), *zpld1a*, *zpld1b* (genes involved in the development of the cupula of the inner ear, which is responsible for balance) and *ptf1a*, *grid2* and *cbln12* (expressed in the zebrafish cerebellum).

Previous studies in cells demonstrated that reduced expression of *NOP56* gene leads to decreased pre-rRNA biogenesis and an increase in apoptotic cells [10]. When *NOP58* expression is reduced in cells, increased expression of *NOP56* and *FBL* can be observed [10]. Similar to these results, our data also indicate impaired rRNA processing and increased apoptosis (mainly in the CNS), together with mRNA overexpression of *nop58* and *fbl,* which seems to be a compensatory mechanism for reduced *nop56* expression. A zebrafish *fbl* mutant develops a similar phenotype with severe neurodegeneration accompanied by eye abnormalities, massive apoptosis, defects in ribosome biogenesis and activity, and impaired S-phase progression [46].

Strong *nop56* mRNA expression was previously localized in the retina, posterior midbrain lamina and the cerebellum of zebrafish embryos [56]. In mice Nop56 protein expression was also found mainly in CNS in Purkinje cells of the cerebellum, motor neurons of the hypoglossal nucleus and spinal cord anterior horn [19]. Increased apoptosis restricted to the CNS was observed, which seems to be in concordance with *nop56* expression mainly in the CNS.

Locomotor defects in *nop56^−/−^* could be caused by absence of cerebellum, reduced numbers of 5-HT-ir spinal cord neurons and defected body innervation. Previous studies linked cerebellar ataxia with the impairment of the serotonergic cerebellar system, where it was used as a target for some experimental treatments [57,58,59]. More recently, it has been seen in other ataxia animal models that treatments with serotonergic agents improved significantly the locomotion [60]. In addition, we observed reduced mRNA expression of *zpld1* genes in *nop56^−/−^*, this reduced expression of *ZPLD1* genes in other species was related to balance dysfunction. In mouse, two spontaneous mutations in *Zpld1* gene resulted in vestibular dysfunction but not auditory dysfunction [61]. In humans, a mutation in *ZPLD1* was found in a patient with balanced translocation and cerebral cavernous malformations [62].

Our results show altered mRNA levels of *tdp-43* and *fus* (which are related to ALS) *in nop56^−/−^*. The relationship between TDP-43, FUS and NOP56 was previously signaled by Miyazaki and colleagues [63] in an ALS mouse model, which shows a progressive reduction in the mRNA levels of *Nop56*, *Tdp-43* and *Fus* in large motor neurons. TDP-43 and FUS participate, as NOP56, in RNA processing pathway. We also analyzed mRNA expression of another gene related to ALS, *c9orf72.* This gene not only is an autophagy regulator, but also participates in the regulation of actin dynamics and axon extension in motor neurons [64]. We found a reduced mRNA expression of *c9orf72* in *nop56*^a12582^ mutants, which correlated with reduced numbers of spinal cord neurons.

*Nop56*^sa12582^ homozygous mutant had a severe cerebellar defect; in fact, they did not develop a cerebellum and had a premature death. In humans, the only known illness that affected *NOP56* gene is SCA36. There are no homozygous patients for SCA36. However, the heterozygous intronic repeated expansion of GGCCTG in *NOP56* causes SCA36, which presented less severe phenotype than our *nop56^−/−^* zebrafish with reduced locomotion, sensory hearing loss and discrete motor neuron impairment. Heterozygous *nop56* fishes had reduced *nop56* mRNA expression in comparison with wild types, but not as dramatically as homozygous *nop56^−/−^* fish. *Nop58* and *fbl* mRNA were also overexpressed in *nop56^+/−^* compared with wild type, while *c9orf72*, *fus*, *cbln12*, *grid2* and *zpld1b* had reduced mRNA expression starting mainly at 5 dpf. All these statistically significant differences of expression compared to the wild type are not as strong as those of the *nop56^−/−^*

Our studies in the brain of *nop56^+/−^*adult fishes of 6 months did not show any macroscopic differences and also we did not observe any alteration in *nop56^+/−^* Purkinje cells at 4 dpf. This does not mean that there are no neuronal differences between nop56^+/−^and adult wild type, especially seeing that genes related to Purkinje cells (*grid2*), granular cells (*cbln12*), to balance (*zpld1b*) and genes that have been related to ALS (*c9orf72* and *fus*) had reduced expression compared to wild type. Future research is necessary to see if there really are differences at the CNS level between adult nop56^+/−^ and adult wild type. Although SCA36 illness is thought to be a gain-of-function disease as ALS, recent studies in ALS signaled a causative combination of factors in which haploinsufficiency had an important role [22,24,25,52,65]. For these reasons, future studies on *nop56^+/−^* older adult fishes are necessary.

Mice injected with a construct containing the SCA36 expansion showed not only RNA foci and DPR inclusions in the brain, but also locomotor defects and loss of Purkinje cells [30]. The generation of overexpression of SCA36 expansion in the zebrafish model facilitates the search for candidate therapies through drug or genetic screens to ameliorate the SCA36 symptoms.

## 5. Conclusions

We provided the first vertebrate *NOP56* loss of function model in which we reported a severe neurodegenerative phenotype mainly characterized by early death, increase of apoptosis, absence of cerebellum, reduced numbers of spinal cord neurons and impaired movement. 

## Figures and Tables

**Figure 1 biomedicines-10-01814-f001:**
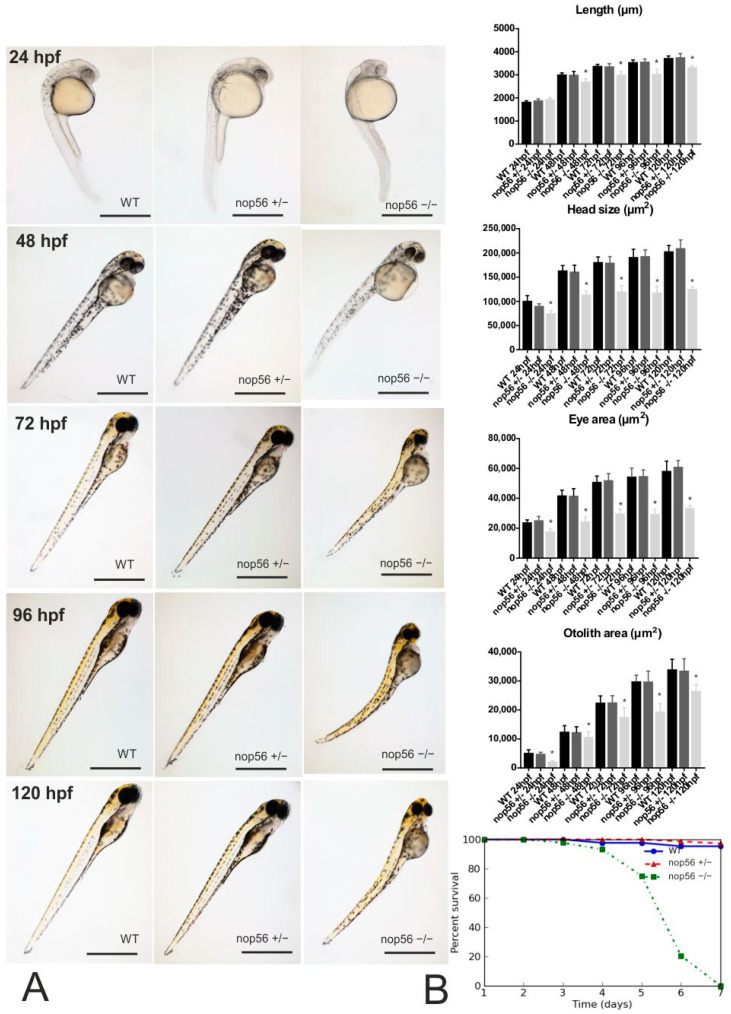
(**A**): Phenotype of wild-type, *nop56*^+/−^ and *nop56*^−/−^ between 24 and 120 hpf. At 24 hpf, *nop56*^−/−^ show statistically significant reduced head size, smaller eyes and smaller. At 48 hpf, statistically significant reduced body length can be also observed in *nop56*^−/−^ embryos, as well as swollen yolk (in 100% of *nop56*^−/−^ embryos) and curved body shape in (40% of the *nop56*^−/−^ embryos). At 72 hpf, in addition to the previous features, there is cardiac edema in 77% of the embryos. In addition, at 96 and 120 hpf, there is total absence of swimming bladder in all *nop56*^−/−^ embryos, curved body shape (in 50% of *nop56*^−/−^ larvae) and a cardiac edema (in 88% of the *nop56*^−/−^ larvae at 96 hpf and 100% at 120 hpf). For these parameters, no statistically significant differences were observed between wild type and *nop56*^+/−^. Statistically significant data in the graphs are indicated with a *. Scale bar: 1000 μm. (**B**) Kaplan–Meier survival graph for wild-type (*n* = 43), *nop56*^+/−^ (*n* = 75) and *nop56*^−/−^ (*n* = 44) zebrafish, revealing significantly different survivals for the *nop56*^−/−^ embryos *p* < 0.0001.

**Figure 2 biomedicines-10-01814-f002:**
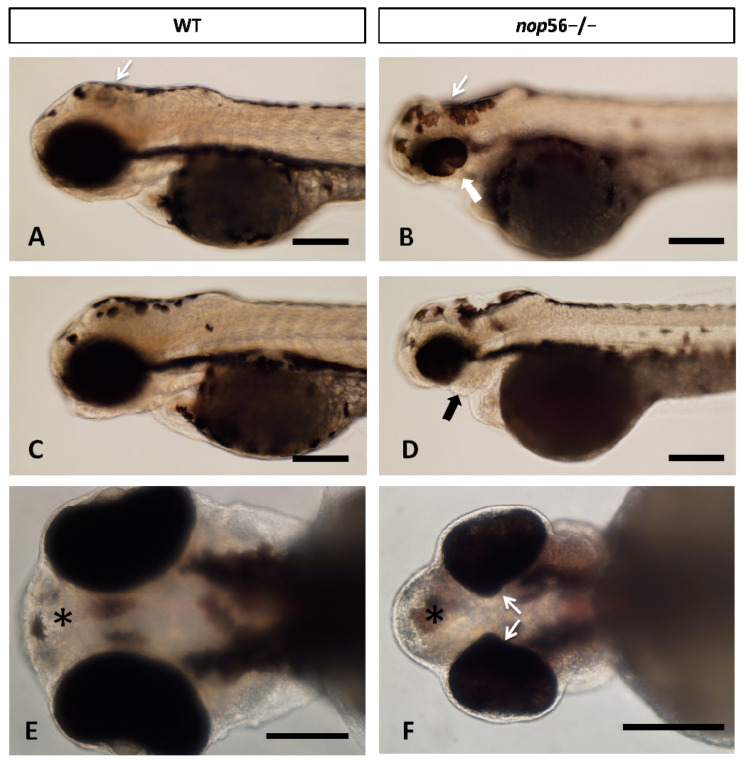
(**A**–**D**) Lateral views of WT and nop56^−/−^ larvae (3.5 dpf), showing coloboma (white thick arrow in (**B**)) and jaw malformation (black arrow in (**D**)) in the homozygous fish. Note also differences in the midbrain and cerebellum (thin white arrows). (**E**,**F**) Ventral view of WT and *nop56*^−/−^ fish showing malformation of the eye (arrows) and forebrain (asterisk). Scale bar: 125 μm.

**Figure 3 biomedicines-10-01814-f003:**
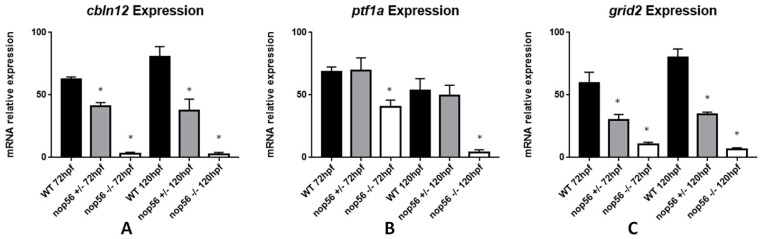
Graphic representation of expression analysis by RT-qPCR. (**A**) *Cbln12* expression is significantly reduced in *nop56*^+/−^ and *nop56*^−/−^ embryos between at 72 and 120 hpf in comparison with wild-type embryos (*p*-value < 0.0001). (**B**) *Ptf1a* expression is significantly reduced in *nop56^−/−^* embryos at 72 and 120 hpf (*p*-value < 0.0001.) (**C**) *Grid2* expression is significantly reduced in *nop56*^+/−^ and *nop56*^−/−^ embryos between at 72 and 120 hpf in comparison with wild-type embryos (*p*-value < 0.0001). Statistically significant data in the graphs is indicated with a *.

**Figure 4 biomedicines-10-01814-f004:**
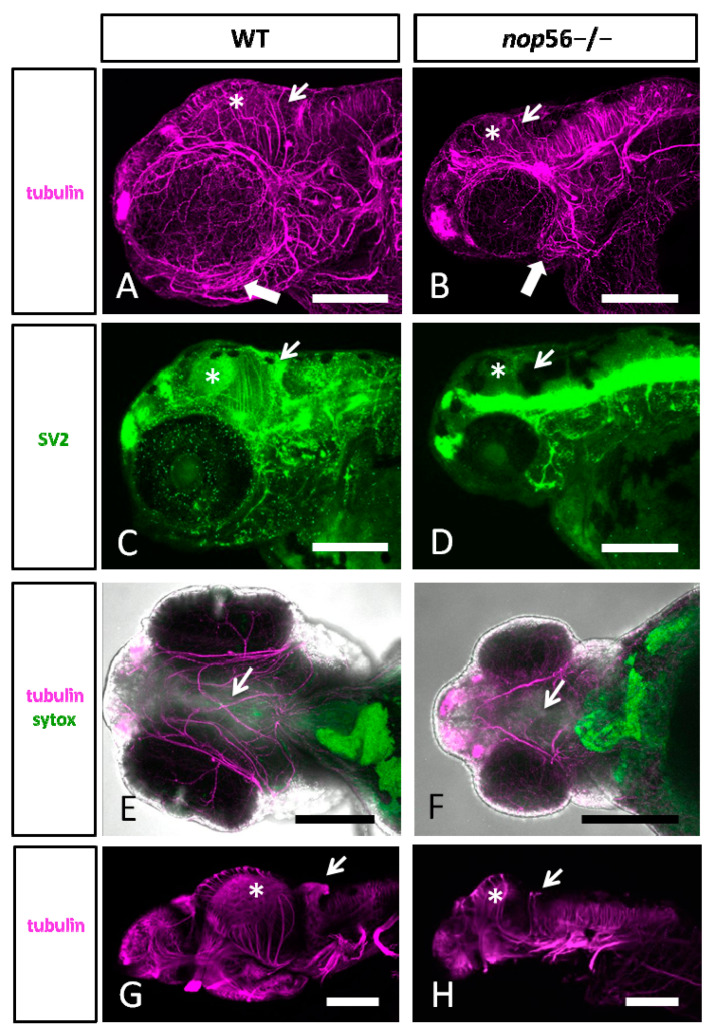
(**A**,**B**) Lateral views of WT (A) and *nop56*^−/−^ (**B**) 3.5 dpf larvae immunostained against α-tubulin (magenta) showing abnormalities in the jaw (thick arrow) and brain, specially in the midbrain (asterisk) and cerebellum (thin arrow). (**C**,**D**) Lateral views of WT (**C**) and *nop56*^−/−^ (**D**) fish immunostained against SV2 (green) showing different distribution in the brain, especially in the midbrain (asterisk) and cerebellum (arrow). (**E**,**F**) Ventral views of WT (**E**) and *nop56*^−/−^ (**F**) showing lack of midline fibers and bundles in the homozygous fish (arrow in (**F**)), compared to WT (arrow in (**E**)). (**G**–**F**) Lateral views of WT (**G**) and *nop56*^−/−^ (**H**) brain showing smaller midbrain in the homozygous fish (asterisk in (**H**)) and absence of labelling in cerebellum (arrow in (**H**)), compared to WT (asterisk and arrows in (**G**)). All images are projections from confocal z-stacks. Scale bars: 125 μm.

**Figure 5 biomedicines-10-01814-f005:**
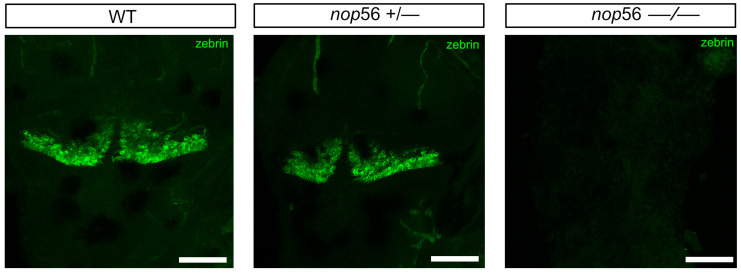
Zebrin II immunofluorescence in 4 dpf zebrafish larvae revealed a complete lack of cerebellar Purkinje cells in *nop56*^−/−^. Dorsal views. Anterior to the top. Scale bar: 150 μm.

**Figure 6 biomedicines-10-01814-f006:**
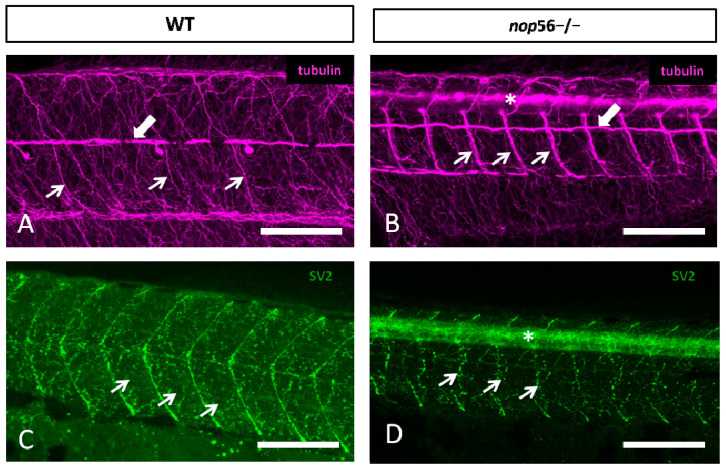
(**A**,**B**) Lateral views of WT (**A**) and *nop56*^−/−^ (**B**) 3.5 dpf larvae showing malformations in the neuromuscular junctions and the innervation of the trunk myomeres. All images are projections from confocal z-stacks. (**A**,**B**) Alpha-tubulin staining (magenta) showing alterations in the distribution of fibers and bundles in the trunk (thin arrows). White thick arrows in (**A**,**B**) point to posterior lateral line nerve. (**C**,**D**) SV2 (green) staining showing alterations in neuromuscular junction (arrows) and myosepta in *nop56*^−/−^ compared to wild type. Asterisk in (**B**,**D**) marks the spinal cord. Scale bar: 150 μm.

**Figure 7 biomedicines-10-01814-f007:**
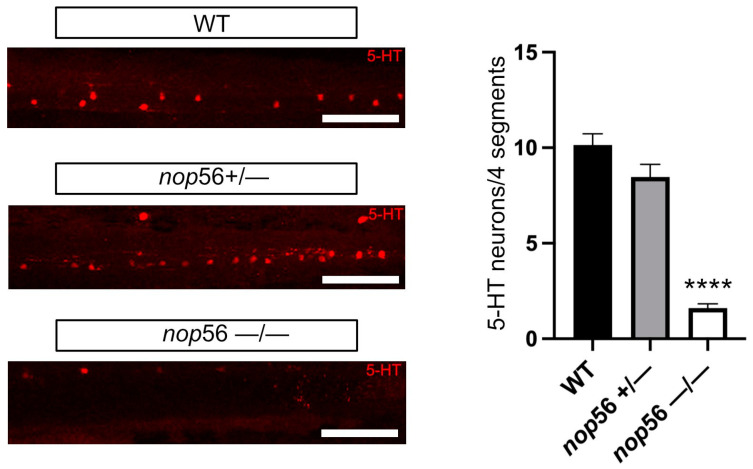
5-HT immunofluorescence in 4 dpf zebrafish larvae revealed significant reduction in the number of 5-HT-ir spinal cord neurons in *nop56*^−/−^ animals. Scale bar: 150 μm. Statistically significant data in the graphs are indicated with ****.

**Figure 8 biomedicines-10-01814-f008:**
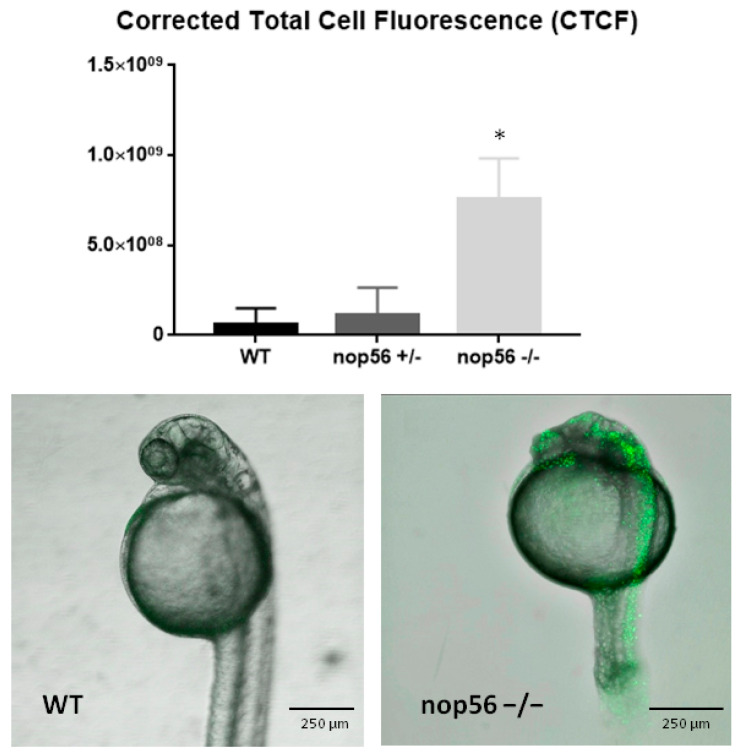
Apoptosis assay in wild type, *nop56*^+/−^ and *nop56*^−/−^. Fluorescence highlights apoptotic cells. *nop56*^−/−^ have a high increase in apoptosis mainly in CNS (*p*-value < 0.0001). Statistically significant data in the graphs are indicated with a *.

**Figure 9 biomedicines-10-01814-f009:**
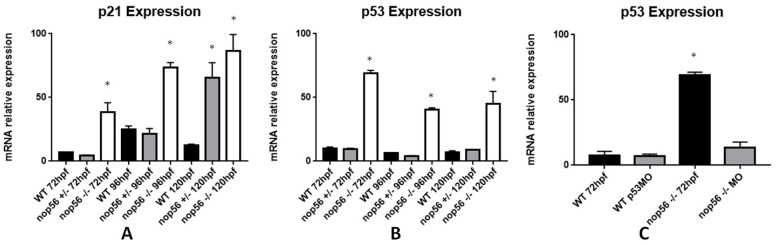
Graphic representation of analysis of mRNA expression by RT-qPCR. (**A**) *p21* expression is significantly higher in *nop56*^−/−^ larvae between 72 and 120 hpf (*p*-value < 0.0001.) (**B**) *p53* expression is significantly higher in *nop56*^−/−^ larvae between 72 and 120 hpf (*p*-value < 0.0001.) (**C**) Microinjection of *p53* morpholino restored the expression in *nop56*^−/−^ embryos to levels similar to wild type but not the malformations (data not shown). Statistically significant data in the graphs are indicated with a *.

**Figure 10 biomedicines-10-01814-f010:**
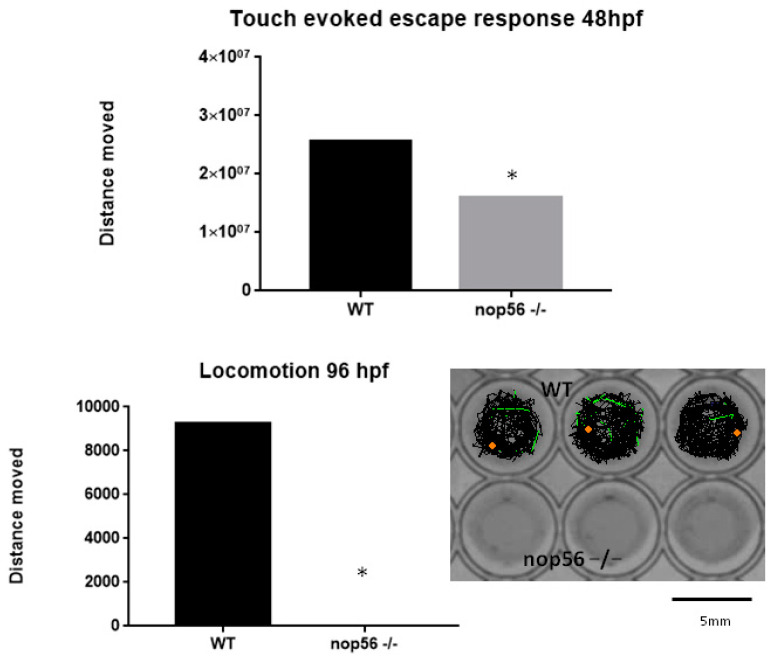
Touch-evoked escape was reduced at 48 hpf in *nop56*^−/−^ embryos and locomotion was absent in nop56^−/−^ larvae at 96 hpf (*p*-value < 0.0001). Statistically significant data in the graphs are indicated with a *.

**Figure 11 biomedicines-10-01814-f011:**
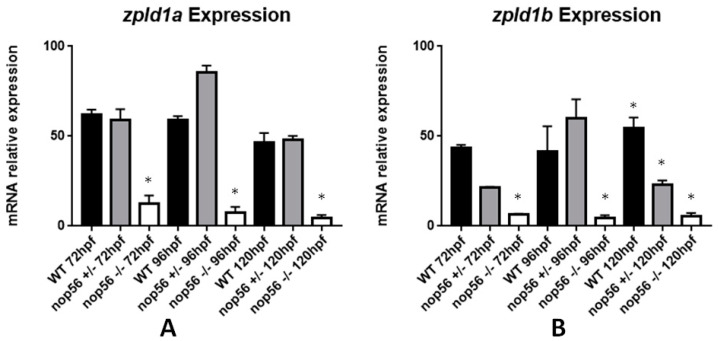
Graphic representation of expression analysis by RT-qPCR. (**A**) *Zpld1a* expression is significantly reduced in *nop56*^−/−^ larvae between 72 and 120 hpf (*p*-value < 0.0001). (**B**) Zpld1a expression is significantly reduced in *nop56*^−/−^ larvae between 72 and 120 hpf (*p*-value < 0.0001) and in *nop56*^+/−^ at 120 hpf. Statistically significant data in the graphs are indicated with a *.

**Figure 12 biomedicines-10-01814-f012:**
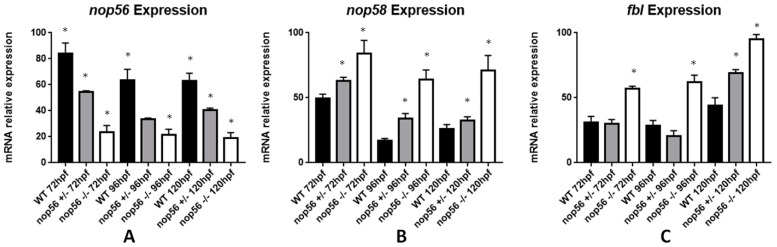
Graphic representation of mRNA expression analysis by RT-qPCR. (**A**) *nop56* expression is significantly different in wild-type, *nop56*^+/−^ and *nop56*^−/−^ larvae between 72 and 120 hpf (*p*-value < 0.0001). (**B**) *nop58* expression is significantly higher in *nop56*^−/−^ and *nop56*^+/−^larvae between 72 and 120 hpf (*p*-value < 0.0001.) (**C**) *fbl* expression is significantly higher in *nop56*^−/−^ larvae between 72 and 120 hpf (*p*-value < 0.0001) and in *nop56*^+/−^ at 120 hpf respect to wild-type larvae (*p*-value < 0.0001). Statistically significant data in the graphs are indicated with a *.

**Figure 13 biomedicines-10-01814-f013:**
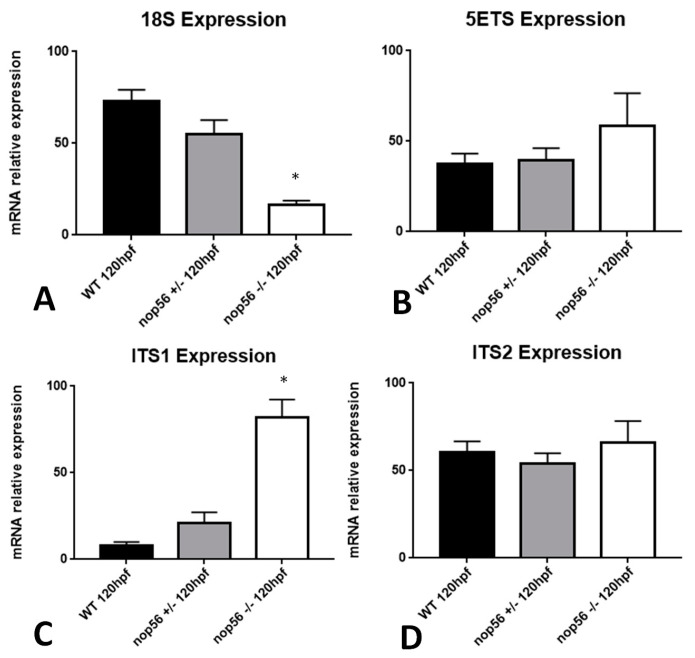
Graphic representation of expression analysis by RT-qPCR. (**A**) 18S rRNA expression is significantly reduced in *nop56*^−/−^ larvae at 120 hpf (*p*-value < 0.0001). (**B**) 5ETS expression does not differ between wild type, *nop56*^−/−^ and *nop56*^+/−^ embryos at 120 hpf (**C**) ITS1 expression is significantly higher in *nop56*^−/−^ larvae between at 120 hpf (*p*-value < 0.0001). (**D**) ITS2 expression does not differ between wild type, *nop56*^−/−^ and *nop56*^+/−^ larvae at 120 hpf. Statistically significant data in the graphs are indicated with a *.

**Figure 14 biomedicines-10-01814-f014:**
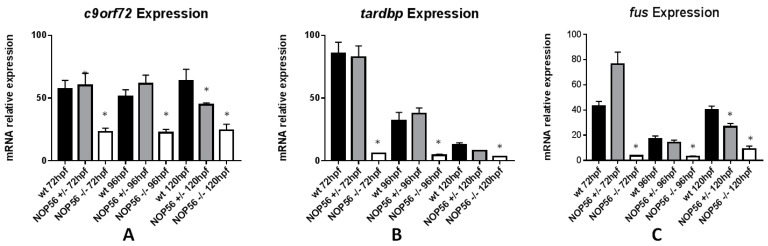
Graphic representation of mRNA expression analysis by RT-qPCR. (**A**) *c9of72* expression is significantly reduced in *nop56*^−/−^ embryos between 72 and 120 hpf (*p*-value < 0.001) and in *nop56^+/−^* at 120 hpf (*p*-value=0.0015). (**B**) *tardbp* expression is significantly reduced in *nop56^−/−^* embryos between 72 and 120 hpf (*p*-value < 0.001). (**C**) *fus* expression is significantly reduced in *nop56^−/−^* embryos between 72 and 120 hpf (*p*-value < 0.0001) and in *nop56^+/−^* embryos at 120 hpf (*p*-value < 0.0001). Statistically significant data in the graphs are indicated with a *.

## Data Availability

The data that support the findings of this study are available from the corresponding author upon reasonable request.

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
