# Peer review of "A nop56 Zebrafish Loss-of-Function Model Exhibits a Severe Neurodegenerative Phenotype"

_biomedicines, 2022, doi:10.3390/biomedicines10081814_

Round 1

Reviewer 1 Report

1. As the manuscript described, both ptf1a and grid2 are markers of purkinje cells. In Figure 3B and 3C, the expression of grid2 is not only significantly reduced in nop56-/- but also in nop56+/- at the stage of 72 and 12 hpf. But there is no description in the text about the significantly reduced grid2 in nop56+/- embryos. Did the nop56+/- embryos exhibit the defective cerebellum (or purkinje cells)? Suggesting to describe that and the physical meaning in the article. 

2. Figure 2B. What is the meaning of the thin arrow?  In the legend, “…in the midbrain and cerebellum (thick white arrows)” …Typing error? Please recheck it.

3. Section 3.4. It is described “Acridine orange…, mainly in the eye, brain, and spinal cord (figure 8). In Figure 8, the live images of WT and nop56-/- were shown in different orientations, which is difficult to distinguish the increased apoptotic areas. It is better to show the nop56-/- embryos in lateral view (the same orientation as the WT).

4. Expression of p53 was dramatically increased in nop56-/- embryos at 72 and 120 hpf. In Figure 9C, the authors knock-downed p53 expression by injecting p53-specific morpholino which resulted in the reduced expression of p53 (expression level even similar to WT). How does the MO-knocking down of P53 protein translation lead to inhibiting p53 gene expression (reduced p53 mRNA level)? The author did not interpret this in the text (or in the discussion section).

5. Figure 11. Expression of zpld1b was also significantly reduced in nop56+/- embryos at 120 hpf, but the authors did not have any description and discussion in the manuscript.

Minor comments:

1. Please add “A” symbol in Figure 1.

2. Please recheck the citing format of the journal’s name (some are shown as full name, some are shown as abbreviation style).

Author Response

We would like to thank the reviewer for her/his suggestions in order to improve the paper’s quality. We have followed all the reviewer’s suggestions.

  1. As the manuscript described, both ptf1a and grid2 are markers of purkinje cells. In Figure 3B and 3C, the expression of grid2 is not only significantly reduced in nop56-/- but also in nop56+/- at the stage of 72 and 12 hpf. But there is no description in the text about the significantly reduced grid2 in nop56+/- embryos. Did the nop56+/- embryos exhibit the defective cerebellum (or purkinje cells)? Suggesting to describe that and the physical meaning in the article. 

We included in the manuscript the phrase: “grid2 and cbln12  expression was observed to be also reduced in nop56+/-   between  72 and 120 hpf in comparison with wild type embryos (p-value < 0.0001) but not as strong as nop56-/- larvae”

As we indicated in the results, nop56+/- embryos did not exhibit defective cerebellum or purkinje cells at 4 dpf, although we did not studied further stages.

In the discussion we explain the results obtained in nop56+/-: “Heterozygous nop56 fishes had reduced nop56 mRNA expression in comparison with wild types, but not as dramatically as homozygous nop56-/- fish. Nop58 and fbl mRNA were also overexpressed in nop56+/- compared with wild type, while c9orf72, fus, cbln12, grid2, zpld1b had reduced mRNA expression starting mainly at 5 dpf. All these statistically significant differences of expression compared to the wild type are not as strong as those of the nop56-/-

“Our studies in the brain of nop56+/-adult fishes of 6 months did not show any macroscopic differences and also we did not observe any alteration in nop56+/- purkinje cells at 4 dpf. This does not mean that there are no neuronal differences between nop56+/-and adult wild type, especially seeing that genes related to purkinje cells (grid2), granular cells (cbln12), to balance (zpld1b) and genes that have been related to ALS (c9orf72 and fus) had reduced expression compared to wild type. Future research would be necessary to see if there really are differences at the CNS level between adult nop56+/- and adult wild type”

  1. Figure 2B. What is the meaning of the thin arrow?  In the legend, “…in the midbrain and cerebellum (thick white arrows)” …Typing error? Please recheck it.

Yes, it was a typing error. The legend would be: “Note also differences in the midbrain and cerebellum (thin white arrows)”

  1. Section 3.4. It is described “Acridine orange…, mainly in the eye, brain, and spinal cord (figure 8). In Figure 8, the live images of WT and nop56-/- were shown in different orientations, which is difficult to distinguish the increased apoptotic areas. It is better to show the nop56-/- embryos in lateral view (the same orientation as the WT).

I tried my best to put the fishes in the same orientation. I changed the WT image to make the figure easier to interpret. Unfortunately, all the images we have of the nop56 -/- embryos are in the same orientation, since due to the increased size of the yolk, it is very difficult to orient them before 48 hpf. In addition, right now we do not have more embryos to repeat the orange acridine experiment.

  1. Expression of p53 was dramatically increased in nop56-/- embryos at 72 and 120 hpf. In Figure 9C, the authors knock-downed p53 expression by injecting p53-specific morpholino which resulted in the reduced expression of p53 (expression level even similar to WT). How does the MO-knocking down of P53 protein translation lead to inhibiting p53 gene expression (reduced p53 mRNA level)? The author did not interpret this in the text (or in the discussion section).

It was added to the result section: The reduction in p53 mRNA concentrations would be explained by the fact that it was observed that sometimes when a translation block morpholino binds to an mRNA, its secondary structure suffer changes, altering the availability of mRNA for nucleotyc degradation [50].

  1. Figure 11. Expression of zpld1b was also significantly reduced in nop56+/- embryos at 120 hpf, but the authors did not have any description and discussion in the manuscript.

We included in the result section the phrase “zpld1b was also significantly reduced in nop56+/- embryos at 120 hpf, but not as strong as npc1-/- expression”

In the discussion we explain the results obtained in nop56+/-: “Heterozygous nop56 fishes had reduced nop56 mRNA expression in comparison with wild types, but not as dramatically as homozygous nop56-/- fish. Nop58 and fbl mRNA were also overexpressed in nop56+/- compared with wild type, while c9orf72, fus, cbln12, grid2, zpld1b had reduced mRNA expression starting mainly at 5 dpf. All these statistically significant differences of expression compared to the wild type are not as strong as those of the nop56-/-

“Our studies in the brain of nop56+/-adult fishes of 6 months did not show any macroscopic differences and also we did not observe any alteration in nop56+/- purkinje cells at 4 dpf. This does not mean that there are no neuronal differences between nop56+/-and adult wild type, especially seeing that genes related to purkinje cells (grid2), granular cells (cbln12), to balance (zpld1b) and genes that have been related to ALS (c9orf72 and fus) had reduced expression compared to wild type. Future research would be necessary to see if there really are differences at the CNS level between adult nop56+/- and adult wild type”

Minor comments:

  1. Please add “A” symbol in Figure 1.

A symbol was added to Figure 1.

  1. Please recheck the citing format of the journal’s name (some are shown as full name, some are shown as abbreviation style).

All the cites were checked and put in the same format.

Reviewer 2 Report

The research manuscript by Dr Quelle-Regaldie et al., entitled “A nop56 zebrafish loss of function model exhibits a severe neurodegenerative phenotype, deals with the characterization of a genetic model of NOP56 loss of function in Zebrafish. They found that nop56 mutants are characterized by severe development impairment, consisting in absence of cerebellum, reduced spinal cord neuron and high levels of apoptosis in the Central Nervous System (CNS). These severe alterations are expected due to the function of the nop56, which is part of the protein complex C/D box small nucleolar ribonucleoprotein (snoRNP), which is responsible for cleavage and modification of precursor ribosomal RNAs (pre-rRNAs) and assembly of the 60S ribosomal subunit. These findings add interesting information to the known effects of intronic expansions of the NOP56 gene, which cause a late onset autosomal dominant ataxia in Vertebrates (“SCA36 expansion”).

These findings indicate that NOP56 is fundamental for the basic development of the motor systems which involve both the efferent and the afferent components (cerebellum, spinal cord neurons, neuromuscular junction, inner ear). NOP56 expression seems to influence the expression of proteins involved in ALS and in SCA(36).

The study is carried out with a solid rational, results are adequately presented and discussed appropriately, with a correct citation of the previous literature.

I have not change to suggest.

Author Response

We would like to thank the reviewer for her/his positive comments.

Round 2

Reviewer 1 Report

I would suggest the authors carefully recheck the reference citing issues.

1. Please recheck the citing format of the journal’s name (some are shown as full name (2, 4, 5, … ), some are shown as abbreviation style (7,13, 25,… )).

2. cited reference #6, "Journal name???"

3. cited reference #22, please recheck the format.

Author Response

We follow your recommendations and reviewed and changed all the reference list